# Evaluation of Antimicrobial Resistance Profiles of Bacteria Isolated from Biofilm in Meat Processing Units

**DOI:** 10.3390/antibiotics12091408

**Published:** 2023-09-05

**Authors:** Alexandra Ban-Cucerzan, Adriana Morar, Emil Tîrziu, Kálmán Imre

**Affiliations:** Department of Animal Production and Veterinary Public Health, Faculty of Veterinary Medicine, University of Life Sciences “King Mihai I” from Timişoara, 300645 Timișoara, Romania; emiltirziu@usvt.ro (E.T.); kalmanimre@usab-tm.ro (K.I.)

**Keywords:** antibiotics, resistance, biofilm, meat

## Abstract

The aim of this study was to assess the hygiene of pork, beef, and poultry carcasses and to determine the phenotypic antibiotic susceptibility of the bacteria embedded in the biofilm formed on the carcasses kept in cooling chambers for at least three days. The level of hygiene was assessed by determining the total aerobic colony count (TACC) and the *Enterobacteriaceae* level in different sampling points of the carcasses, along with the detection of *E. coli* and *Pseudomonas* spp. embedded in the biofilm. Furthermore, the *E. coli* and *Pseudomonas* spp. isolates were tested for antimicrobial resistance profiles. A total of 130 samples collected from pork, beef, and poultry from processing units were analyzed to determine the total aerobic colony count as well as to measure the level of *Enterobacteriaceae* found on the carcasses. The antimicrobial susceptibility of 44 *Escherichia coli* and eight *Pseudomonas* spp. strains isolated from the carcasses were assessed using the Vitek 2 system using two different cards. Overall, the regulatory limits for the TACC were exceeded in 7.6% of the samples, and 65% of the samples exceeded the regulatory limits for *Enterobacteriaceae* levels. The antimicrobial susceptibility tests of the *E. coli* isolates analyzed with the AST-GN27 card revealed the highest resistance to be that towards ampicillin (76.1%), followed by cefazolin (71.4%), amoxicillin/clavulanic acid (61.9%), nitrofurantoin (52.3%), cefoxitin (47.6%), tetracycline (38.1%), piperacillin, norfloxacin (19%), trimethoprim–sulfamethoxazole (11.9%), cefotaxime (9.5%), ceftazidime, cefazolin, amikacin, gentamicin, and ciprofloxacin (4.7%). However, all of the isolates were sensitive to piperacillin-tazobactam and imipenem. Thirty-two (61.5%; 95% CI 47.9–73.5) out of fifty-two isolates exhibited multidrug resistance, resulting in the expression of 10 resistance profiles. The findings of this study highlight serious hygienic and sanitary deficiencies within the meat processing units and demonstrate that the resulting meat can harbor Multidrug-resistant *Escherichia coli* and *Pseudomonas* spp., both of which pose a serious public health risk. However, further research with a larger number of samples is required to reach thorough results.

## 1. Introduction

Irrational antimicrobial usage in farm animals is the main factor in the development of Multidrug-resistant bacteria in meat and meat products [1]. The antimicrobial resistance phenomenon (AMR) is considered a global public health concern, leading to Multidrug-resistant bacteria that can cause life-threatening diseases [2].

The costs of the presence and spread of the MDR bacteria are very high, as it directly affects productivity through prolonged treatments and the need for more expensive and intensive care. Also, standard procedures such as surgery, cancer chemotherapy, or organ transplantation might increase the risk [3].

The European Union’s (EU) Farm to Fork strategy aims to reduce the total antimicrobial usage in farm animals, and in 2021, the EU Member States will achieve an 18% reduction in total sales. The initiative targets a 50% reduction in antimicrobial sales by 2030 [4].

The AMR phenomenon is also enhanced by the capacity to form a biofilm. Biofilms are intricate microbial ecosystems made up of one or more species embedded in an extracellular matrix of different compositions [5,6]. Current research shows that the bacteria embedded in the biofilm are not exposed to the corresponding effective dose of the antimicrobial agent because the biofilm matrix limits drug penetration, resulting in the emergence of MDR. There are numerous foodborne pathogens (e.g., *Bacillus cereus*, *Escherichia coli*, *Listeria monocytogenes*, *Salmonella enterica*, *Staphylococcus aureus*, *Campylobacter jejuni*, *Pseudomonas* spp., etc.) that can build biofilms. Most of the biofilms are formed in parts of the meat factories that are challenging to effectively sanitize and, over time, can also exhibit resistance to the utilized sanitizers [7,8,9,10,11]. *P. aeruginosa* is an opportunistic bacteria that may colonize and persist in nearly any environment. The infections pose a serious medical problem because of the easily acquired antimicrobial resistance of the bacteria [12,13].

The formation and quantity of the biofilm can also be influenced by the exposure temperature. Research in this field showed that lower temperatures (22 °C) will influence a higher production of biofilm compared to temperatures above 30 °C. This can be due to the production mechanism of specific components (e.g., aggregative fimbriae of *Salmonella* spp.) at specific temperatures, which for most of the bacteria are below 30 °C. The ability to form biofilm at low temperatures represents an advantage for the pathogenic bacteria that will persist in the environment and will continue to raise a serious health concern for the food industry [14].

The presence of *E. coli* is widely tested through European Food Safety Surveillance Programs aimed to assess food hygiene since the bacteria is recognized as an important indicator of fecal contamination [15]. In terms of AMR, *E. coli*-type bacteria are used as sentinels in humans, as well as animals, since they can survive in a variety of conditions and rapidly acquire antimicrobial resistance that can be passed to other bacteria via plasmid mediation. Furthermore, the MDR *E. coli* from the meat can represent a contamination source for humans that come into direct contact with uncooked or undercooked meat [1].

Contamination in meat processing can occur during several stages of the processing line. The carcasses in the meat processing units could be contaminated during transport from the slaughterhouse or during handling within the facility [16]. The Commission Regulation (EC) No. 1441/2007 establishes a maximum contamination level of 5.0 log CFU/cm^2^ for the total aerobic colony count (TACC) and 2.5 log CFU/cm^2^ for the *Enterobacteriaceae* isolated from beef carcasses prior to refrigeration, ensuring that the meat is obtained in hygienic conditions [17]. The contamination level of the carcasses in meat processing units should not exceed the established limits, especially due to the fact that the meat has already been subjected to the chilling stage in the slaughterhouse, which restricts bacterial development.

The aim of this study was to assess the hygiene status of pork, beef, and poultry carcasses and to determine the antimicrobial resistance profiles of the bacteria embedded in the biofilm formed on the carcasses. The hygiene status was assessed by determining the total aerobic colony count (TACC) and the *Enterobacteriaceae* level at different points of the carcasses, along with the detection of *E. coli* and *Pseudomonas* spp. embedded in the biofilm. Furthermore, the *E. coli* and *Pseudomonas* spp. isolates were tested for antimicrobial resistance profiles, and correlations between similar resistance patterns among the three species were assessed. The current study provides important information regarding the hygiene levels of meat sold in Romania. At the present moment, there are few studies that provide data on the occurrence of MDR *E. coli* strains in meat and meat products sold in Romania [18,19,20,21,22]; however, most of the studies focused on identifying the MDR in live animals.

## 2. Results

### 2.1. Total Aerobic Colony Count Level

Microbiological analysis of 130 samples showed satisfactory TACC levels (<5.0 log CFU/cm^2^) for 120 (92.3%; 95% CI 86.4–95.7) samples and unsatisfactory levels for a significantly lower (*p* < 0.001) number of samples, more specifically 10 (88.4%; 95% CI 4.2–13.5) samples (>5.0 log CFU/cm^2^). Overall, 2 (10%; 95% CI 27.9–30.1) out of 20 sampling points exceeded the daily average for the TACC according to the Commission Regulation (EC) No 1441/2007 [17].

The level of TACC on the pork carcasses ranged from 136 ± 2.1 CFU/cm^2^ to 287,150 ± 1207.5 CFU/cm^2^, with the highest level of contamination being on the inner thigh area followed by the axillary area and the ventral abdominal area (Figure 1). The lowest contamination level was obtained in the neck area. The overall TACC level for the pork carcasses was established at 3.8 × 10^4^ and was considered unsatisfactory.

The TACC level was increased on the ventral parts of the pork carcasses compared with the dorsal parts. For the dorsal parts, the average TACC level was 1644.6 ± 113.3 CFU/cm^2^, compared with the ventral parts, where the level was 75,774 ± 10,561.1 CFU/cm^2^.

For the beef carcasses, the level of contamination ranged from 160.1 ± 54.2 CFU/cm^2^ to 233,700.4 ± 5290.1 CFU/cm^2^, with the lowest level of contamination obtained in the ribcage area, and the highest level reported in the ventral thoracoabdominal region, followed by the chest and inner thigh area (Figure 1). The overall TACC values were 3.1 × 10^4^ CFU/cm^2^ and were considered unsatisfactory.

Similar to the pork carcasses, the TACC level for the beef carcasses was higher on the ventral parts of the carcasses, with an average of 73,083.5 ± 18,115.4 CFU/cm^2^, in contrast to the dorsal parts, where the level was 2141.3 ± 150.9 CFU/cm^2^.

The poultry carcasses had the following average contamination level: 177.4 ± 50.3 CFU/cm^2^ in the axillary region, 340.5 ± 71.1 CFU/cm^2^ in the back area, and 795.1 ± 224.0 CFU/cm^2^ in the chest area (Figure 1). The overall TACC level was 4.3 × 10^2^ CFU/cm^2^ and was considered satisfactory.

### 2.2. Enterobacteriaceae Level

The *Enterobacteriaceae* level of contamination was <1.5 log for 45 (34.6%; 95% CI 26.9–43.1) of the samples, while a significantly higher (*p* < 0.004) number of samples (*n* = 85) indicated a higher level of *Enterobacteriaceae* and were classified as unsatisfactory according to Regulation (EC) No 1441/2007 [17] that establishes a limit of <3.0 log CFU/cm^2^ for pork carcasses and <2.5 CFU/cm^2^ for beef carcasses. The detailed distribution of TACC and *Enterobacteriaceae* on meat carcass samples according to the carcass origin (beef, pork, or poultry) is presented in Table 1.

In pork carcasses, the *Enterobacteriaceae* level exceeded the maximal reference value in five sampling points, namely the inner thigh, the coccygeal area, the ventral abdominal area, the lumbar region, and the sternum.

The *Enterobacteriaceae* level in beef carcasses exceeded the maximal reference value in three sampling points: the inner thigh, ventral thoracoabdominal region, and gluteal muscle area. In poultry carcasses, all of the sampling points exceeded the regulatory limits.

The microbiological analysis of 130 samples (beef, pork, and poultry) showed that 30.7% were contaminated with *E. coli*, with a distribution of 60% on pork carcasses and 22.2% on poultry carcasses. These differences were statistically significant (*p* < 0.01). Overall, 95 (73%; 95% CI 64.8–79.7) of the carcasses revealed values <1 CFU/cm^2^ of *E. coli*.

*Serratia* spp. (*S. liquefaciens* and *S. marcescens*) was present in 40 (30.7%; 95% CI 23.4–39.1) samples and had a distribution of 25% on beef carcasses, 37.5% on pork carcasses, and 37.5% on poultry carcasses. These differences were not statistically significant (*p* > 0.05).

*Enterobacter* spp. (*E. cloacae* and *E. amnigenus*) was isolated from 25 (19.23%; 95% CI 13.3–26.8) samples. *E. cloacae* was isolated from pork (15/20) and beef (5/20) carcasses, and *E. amnigenus* was isolated only from pork carcasses.

*Pseudomonas* spp. (*P. aeruginosa* and *P. putida*) was isolated from 20 (15.3%; 95% CI 10.1–22.5) samples. *P. aeruginosa* was isolated from 6/10 pork samples and from 7/10 beef samples, whereas *P. putida* was isolated from 4/10 pork samples and from 3/10 beef samples.

*Pseudomonas* spp. on the pork carcasses was isolated from six sampling points, namely the neck area, lumbar area, chest, sternum, ventral abdominal area, and intercostal area, whereas on the beef carcasses, it was isolated from one sampling point (ventral thoracoabdominal area).

No significant differences (*p* > 0.05) were obtained regarding the contamination level on different origin carcasses. The results regarding the isolated strains and contamination levels are reported in Table 2.

### 2.3. Identification and Antimicrobial Resistance Profiles

The Vitek 2 system identified the 52 chosen colonies as *E. coli* (*n* = 44), *P. aeruginosa* (*n* = 4), and *P. putida* (*n* = 4). Table 3 provides details related to the strains identified with the Vitek 2 system depending on the source of origin.

Table 4 provides information regarding the antimicrobial resistance profile of 42 *E. coli* strains tested with the AST-GN27 card.

Table 5 provides information regarding the strains (*E. coli* (*n* = 2), *P. aeruginosa* (*n* = 4), and *P. putida* (*n* = 4)) that were tested with the AST-N093 card.

*E. coli* isolates (n = 42) processed with the AST-GN27 card expressed resistance in descending order to AMP (76.1%), CFZ (71.4%), AMC (61.9%), NIT (52.3%), CXT (47.6%), TET (38.1%), PIP, NOR (19%), STX (11.9%), CTX (9.5%), CAZ, CPM, AMK, GEN, and CIP (4.7%); however, all of the isolates were susceptible to TZP and IMP. Additionally, the remaining *E. coli* isolates (*n* = 2), as determined by the AST-N093 card, were resistant to PIP, TIC, TIM, CAZ, CMP, PEF, SXT, and AZM (100%) and susceptible to TZP, IMP, MEM, AMK, GEN, ISP, TOB, CIP, MIN, and COL.

The *P. aeruginosa* isolates (*n* = 4) processes with the AST-N093 card expressed resistance to TIM, TIC, and AZM (100%), whereas *P. putida* strains expressed resistance to TIM, TIC, AZM (100%), and TZP (50%).

Thirty-two (61.5%; 95% CI 47.9–73.5) out of 52 isolates exhibited MDR, resulting in the expression of 10 resistance profiles (Table 6), while two (3%; 95%–CI 1–12) isolates were susceptible to all tested antibiotics. The MDR samples had the following distribution: 18 (34.6%; 95% CI 23.1–48.2) samples were isolated from pork carcasses, 10 (19%; 95% CI 10.8–31.9) from poultry carcasses, and four (7%; 95% CI 3–18.1) from beef carcasses. Three similar patterns of antibiotic resistance were observed: one of the patterns (AMP, AMC, CFZ, CXT, NIT) was identified in all three species (pork, beef, and poultry); one was similar for beef and pork carcasses (AMP, AMC, CFZ, CXT, TET, NIT); and one was similar for pork and poultry carcasses (AMP, PIP, NOR, TET). Statistical differences (*p* < 0.05) were obtained regarding the distribution of the MDR on the poultry carcasses (*n* = 45 samples) and (*p* < 0.007) on the beef carcasses (*n* = 35 samples). Statistical differences were also obtained regarding similar expression patterns of antimicrobial resistance among the tested isolates and their isolation sources (*p* < 0.005).

## 3. Discussions

The average meat consumption (kg/person per month) in Romania in 2020 was 0.3 for beef, 1.3 for pork, 1.6 for poultry, and 1.4 for meat products. The main sources of supply for the population are the local processing units and local stores, which are also supplied by the local processing units [23].

The current study evaluated the hygienic conditions of beef, pork, and poultry carcasses during the storage period in three meat processing units from Timiș County and highlighted serious deficiencies reflected by the high number of samples (65%) that exceeded the acceptable level of *Enterobacteriaceae*. These high levels of contamination are generally attributed to leakage of intestinal contents, cross-contamination from the hind part of the body, and inappropriate handling and/or storage conditions [24]. The results show that appropriate hygiene measures were not successfully implemented.

The sampling points with excessive *Enterobacteriaceae* levels reflect the lack of good practices during the slaughtering process and inappropriate transport and storage conditions that would normally reduce the multiplication of the *Enterobacteriaceae* [15], especially in poultry carcasses, where all of the analyzed samples registered values above the regulatory limit.

Taking into consideration the high level of *Enterobacteriaceae* that was identified on all the analyzed carcasses, a serious public health risk arises. Although the presence of *Salmonella* spp. has not been investigated in this study, the high level of *Enterobacteriaceae* could indicate a dangerous contamination with pathogenic bacteria like *Salmonella* spp.

The presence of *Serratia* spp. is detrimental to beef, pork, and poultry meat quality as they can grow inside the package during distribution and can cause rapid spoilage even under low-temperature conditions, a great loss of valuable protein, and hazardous effects on human health [25].

The two identified species, *S. liquefaciens* and *S. marcescens*, are considered the most pathogenic species of the genus and can cause infections in immunocompromised individuals. In the present study, 30.7% of the samples (pork, beef, and poultry) were contaminated with *Serratia* spp. The findings suggest a lack of hygiene during the slaughtering process. Therefore, precautionary measures should be taken to limit the contamination at the slaughterhouse: sterilization of the knife in hot water, excision of the parts with visible contamination, and washing the carcass with drinking water.

The increased level of *Pseudomonas* spp. contamination of the carcass surface identified in the processing units is the result of contamination during transport and bacterial multiplication at low temperatures. It can also occur when the carcasses are stored for an extended period of time. The use of contaminated raw material in the production process results in highly contaminated products that require heat treatment at higher temperatures and prolonged periods of exposure; in addition, the preservation capacity is greatly reduced [26].

The TACC levels were considered acceptable for 92.3% of the collected samples; the highest level of contamination was identified on the ventral abdominal area of the pork carcasses (5.4 log CFU/cm^2^). Comparatively, the ventral thoracoabdominal area of beef carcasses had a level of 5.3 log CFU/cm^2^.

Overall, TACC levels were higher in pork carcasses than in beef and poultry carcasses. The average was 2.6 log CFU/cm^2^ for poultry carcasses, 4.5 log CFU/cm^2^ for beef carcasses, and 4.6 log CFU/cm^2^ for pork carcasses.

In the present study, the analyzed *E. coli* isolates expressed the highest resistance to ampicillin (AMP) and amoxicillin/clavulanic acid (AMC), which are commonly used in our country. The high resistance rates for β-lactam antimicrobial agents among the 44 *E. coli* isolates are similar to other studies [27,28] and are primarily mediated by β-lactamases, which hydrolyze the β-lactam ring and thus inactivate the antibiotic [29].

The EFSA 2021 report on *E. coli* antibiotic-resistant strains isolated from pork and beef revealed increasing trends for ampicillin, ciprofloxacin, cefotaxime, and tetracycline resistance [30].

Although cephalosporin resistance is defined as stable and modest in the same publication [30], in the present study, cephalosporin resistance reached up to 57%, and 81% of the MDR patterns contained a cephalosporin component.

The association between tetracycline, ampicillin, sulfamethoxazole, and trimethoprim as the most frequently represented patterns in European countries is mentioned in the EFSA 2021 report [30] and is similar to the patterns shown in the current study.

Although nitrofurantoin resistance is rare due to its ability to have few bacterial targets [31], in the present study, 52.3% of the isolated strains expressed resistance to the antimicrobial agent.

In the current study, some of the isolates (4.7%) expressed resistance to ciprofloxacin, although other studies concluded that ciprofloxacin has the potential to control biofilm formation produced by *E. coli* and *P. aeruginosa*, limiting the development of resistant bacteria [32,33].

The analyzed isolates were susceptible to imipenem and meropenem; similar findings were reported in the Czech Republic [34], Italy [1], Belgium [35], and also in the EFSA 2021 report [30].

This susceptibility might be due to the fact that meropenem is also resistant to the majority of beta-lactamases (penicillinases, cephalosporinases, and even metallo-beta-lactamases like NDH-metallo-beta-lactamases), which is another reason it has not been misused. Other studies have identified *E. coli* isolates that are up to 10% resistant to meropenem in carcasses and up to 22% resistant in food [36].

The common antimicrobial resistance patterns (AMP, AMC, CFZ, CXT, and NIT) expressed by *E. coli* isolated from pork, beef, and poultry were identified in only one of the processing units. This finding could indicate the formation of a biofilm matrix that contaminates the meat in the processing unit. The antimicrobial resistance profiles identified in the present study suggest that the antibiotics are not used rationally based on an antibiogram, allowing the formation of Multidrug-resistant bacteria that express resistance to up to five classes of antibiotics.

## 4. Materials and Methods

### 4.1. Sample Collection and Bacterial Isolation

Samples were taken from three meat processing units in Timiș County, Romania. Each was sampled during a single visit. Scrapings and swabs were used to collect samples from the surface of carcasses that had been kept in cooling chambers for at least three days. The aseptically collected samples were each packaged in a polyethylene bag (Sterile Ziploc^®^, Sklar Instruments, West Chester, PA, USA), labeled with the sampling date and the area, and delivered under refrigeration conditions (4 °C) to the Food Hygiene and Microbiological Risk Assessment Laboratory, Faculty of Veterinary Medicine, Timisoara [37].

Twenty samples were collected for biofilm screening by scraping the surfaces of the carcasses using sterile surgical blades (SMI, Steinerberg, Belgium) that were subsequently placed in sterile tubes (Deltalabs, Barcelona, Spain).

For the pork carcasses, 10 sampling points were chosen, namely: the masseter muscles, neck area, chest, sternum, intercostal region, axillary region, ventral abdominal area, lumbar region, coccygeal area, and inner thigh area.

For the beef carcasses, seven sampling points were established, namely: the neck area, chest, rib cage area, thoracoabdominal region (dorsal and ventral), inner thigh area, and gluteal muscles area.

For the poultry carcasses, the samples were collected from three sampling points: the chest, axillary region, and back area. The sampling points were adapted from the protocols presented by Barros et al. [38]. The 20 resulting samples were stained with acridine orange (Sigma-Aldrich, Darmstadt, Germany) for one minute, with the dehydrated samples being treated with Hanks modified solution (without D glucose and phenol red) and fixed in ethanol 96% for two minutes before being placed on the slide. A Leica DM 2500 epifluorescent microscope (Leica Microsystems, Wetzlar, Germany) with Leica EL 6000 external UV light source was used to visualize the biofilm. The machine was calibrated at a wavelength of 488 nm, and the visualization was performed with 63× and 100× immersion objectives.

Sample collection for the microbiological testing was performed from a surface of 100 cm^2^, using sterile swabs (Deltalabs, Barcelona, Spain) immersed in physiological peptone solution (PBS, ThermoFisher Scientific, Canada) [38]. The sampling points were identical to the points established for biofilm detection. The number of analyzed carcasses was higher, namely five pork carcasses (50 samples), five beef carcasses (35 samples), and 15 poultry carcasses (45 samples), resulting in a total of 130 swab samples. The sampling protocol was conducted according to ISO 17604/2015 [39].

In the laboratory, the swab samples were immersed in 9 mL of preheated (45 °C) peptone-buffered solution (PBS; pH = 7.5 ± 0.1). Next, serial dilutions of up to 10^−3^ (1:1000) in sterile 0.5% peptone water were prepared; subsequently, 1 mL from each dilution was transferred into a sterile Petri dish (in duplicate) with specific isolation media.

The total aerobic colony count (TACC) was assessed according to ISO 4833/2003 [40] and Commission Regulation (EC) No 1441/2007 using a PCA culture medium and an incubation period of 72 h at 30 °C [17].

The *Enterobacteriaceae* level was assessed according to SR ISO 21528-2/2007 [41]. The samples were cultured on VRBG agar (violet, red, bile, and glucose agar; Biokar Diagnostics, Allone, France) at 37 °C for 24 h. Specific red and violet colonies were subjected to oxidase production and glucose fermentation tests.

According to the regulatory standards established by Commission Regulation (EC) No. 1441/2007 [17] for *Enterobacteriaceae* and TACC, the results were categorized into the following three categories: (1) satisfactory (if the daily mean Log is m, an acceptable level of contamination); (2) acceptable (if it is between m and M, a dangerous level of contamination); or (3) unsatisfactory (if the daily mean is M). The m and M corresponded to 3.5 and 5.0 log CFU/cm^2^ for TACC and to 1.5 and 2.5 log CFU/cm^2^ for *Enterobacteriaceae*, respectively.

The identification of *E. coli* was carried out according to ISO 16649-2/2001 [42] on a chromogenic selective medium consisting of tryptone bile with X-glucuronide (TBX) agar (Oxoid, Basingstoke, Hampshire, UK) at 37 °C, for 4 h and then at 43.5 °C for 24 h. *Pseudomonas* spp. were isolated on Cetrimid agar (Biokar Diagnostics, Allone, France) at a temperature of 37 °C for 18–24 h according to ISO 13720:2010 [43].

### 4.2. Antimicrobial Susceptibility Testing

Before each experiment, the strains were streaked on BHI agar (Brain Heart Infusion, Oxoid; Biokar Diagnostics, Allone, France) plates, and they were subsequently incubated at 37 °C for 24 h.

A total of 52 (pork (*n* = 32), beef (*n* = 10), and poultry (*n* = 10)) colonies were chosen for biochemical characterization and antimicrobial testing. Species were determined by using the automated compact system Vitek2 (bioMérieux, Marcy l’Etoile, France) to analyze all of the biochemical characteristics of the isolates. The reference strains *P. aeruginosa* ATCC 27853 and *E. coli* ATCC 25922 were used as quality controls. The stock cultures were stored at −80 °C in glycerol, and BHI (Brain Heart Infusion, Oxoid; Biokar Diagnostics, Allone, France) and were subsequently incubated at 37 °C for 24 h. The antimicrobial susceptibility pattern of the isolated strains was also determined using the Vitek2 system. In this regard, two different Gram-negative specific cards, namely, AST-GN27 and AST-N093, were applied to monitor the antimicrobial resistance profiles of the isolates. The used cards included a total of 27 antimicrobial substances, from 11 classes, as follows: β-lactams—ampicillin [AMP 2–32 µg/mL], amoxicillin/clavulanic acid [AMC 1–32 µg/mL], piperacillin [PIP 4–128 µg/mL], piperacillin-tazobactam [TZP 1–32 µg/mL], ticarcillin [TIC], ticarcillin/clavulanic acid [TIM]; cephalosporins– cefazolin [CFZ 4–64 µg/mL], cefoxitin [CXT 4–64 µg/mL], cefotaxime [CTX 1–64 µg/mL], ceftazidime [CAZ 1–64 µg/mL], cefepime [CPM 1–64 µg/mL]; carbapenems—imipenem [IPM 0.25–16 µg/mL], meropenem [MEM]; aminoglycosides–amikacin [AMK 2–64 µg/mL], gentamicin [GEN 1–16 µg/mL], isepamicin [ISP], tobramycin [TOB]; fluoroquinolones–ciprofloxacin [CIP 0.25–4 µg/mL], norfloxacin [NOR 6–512 µg/mL], pefloxacin [PEF]; tetracyclines-tetracycline [TET 1–16 µg/mL], minocycline [MIN]; nitrofuran derivative–nitrofurantoin [NIT 0.5–16 µg/mL]; sulfonamides–trimethoprim-sulfamethoxazole [SXT 20–320 µg/mL]; monobactams—aztreonam [AZM]; polymyxins—colistin [COL]. The isolates were categorized by the Vitek2 equipment as susceptible, intermediate, or resistant to the tested antimicrobials. The isolates that expressed an intermediate resistance level to a certain drug were considered resistant [44,45]. The multidrug-resistant (MDR) isolates were those that proved resistant to at least one antimicrobial agent across three or more antimicrobial classes [44,46].

### 4.3. Statistical Analysis

Pearson’s chi-square (ꭓ^2^) test was used to statistically determine distributional variations according to the origin of the sample. For *p* < 0.05, the findings were considered statistically significant.

## 5. Conclusions

The current study found that all three meat processing units (for pigs, cattle, and poultry) had identical circumstances for the formation of biofilms. High levels of *Enterobacteriaceae* and biofilm were present on the carcass after the third day of storage. The isolated bacteria were MDR bacteria, and their presence represents a danger for the consumer, especially if the meat is consumed raw or undercooked. The current study provides background knowledge on the antimicrobial resistance phenotypes of *E. coli* and *P. aeruginosa* strains in the screened area. This study reveals the need to find solutions to reduce the development of biofilm on the stored carcasses, along with improving hygienic conditions in the processing units and reducing antimicrobial treatments to prevent the development of MDR bacteria. To better understand the antimicrobial resistance phenomenon of significant foodborne pathogens in Romania, additional studies using molecular tools are still required to look for antibiotic-resistance genes and mobile genetic elements that are responsible for genetic material transfer between the bacteria embedded in the biofilm.

## Figures and Tables

**Figure 1 antibiotics-12-01408-f001:**
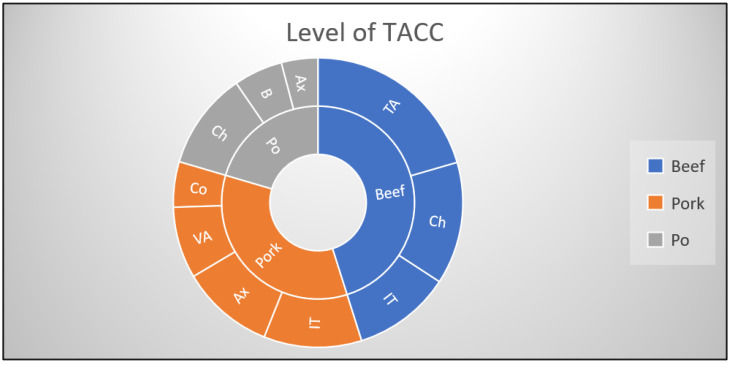
Level of TACC contamination of the investigated carcasses. Legend: TACC—total aerobic colony count, Po—poultry, Ch—chest, IT—inner thigh, TA—thoracoabdominal region, VA—ventral abdominal area, Ax—axillary region, Co—coccygeal area, B—back area.

**Table 1 antibiotics-12-01408-t001:** Meat carcass contamination level assessment using the total aerobic colony count and the *Enterobacteriaceae* level.

Origin of Carcass	No. and (%) of Samples with Different Levels of TACC (log CFU/cm^2^)		No. and (%) of Samples with Different Levels of *Enterobacteriaceae* (log CFU/cm^2^)		
	Below the Regulatory Limits	Above the Regulatory Limits	Below the Regulatory Limits	Above the Regulatory Limits [17]	
	<5.0 log CFU/cm^2^	>5.0 log CFU/cm^2^	<1.5 log CFU/cm^2^	>1.5 log <2.5 log CFU/cm^2^	>2.5 log CFU/cm^2^
Beef (*n* = 35)	30 (85.7)	5 (14.2)	20 (57.1)	10 (28.5)	5 (14.2)
Pork (*n* = 50)	45 (90)	5 (10)	25 (50)	5 (10)	20 (40)
Poultry (*n* = 45)	45 (100)	-	-	15 (30)	30 (60)
TOTAL (*n* = 130)	120 (92.3)	10 (7.6)	45 (34.6)	30 (23)	55 (42.3)

Legend: No—number of samples; *n*—number of samples according to the origin of the carcass; TACC—total aerobic colony count; CFU—colony forming unit.

**Table 2 antibiotics-12-01408-t002:** Isolation of strains and contamination levels of the analyzed meat.

Origin of Carcass	No. of Samples Containing			
	*E. coli* (%)	*Enterobacter* (%)	*Serratia* (%)	*Pseudomonas* (%)
Beef (*n* = 35)	-	5 (14.2)	10 (28.5)	10 (28.5)
Pork (*n* = 50)	30 (60)	20 (40)	15 (30)	10 (20)
Poultry (*n* = 45)	10 (22.2)	-	15 (33.3)	-
TOTAL (*n* = 130)	40 (30.7)	25 (19.2)	40 (30.7)	20 (15.3)

Legend: *n*—number of samples according to the origin of the carcass.

**Table 3 antibiotics-12-01408-t003:** Identified strains depending on the source of origin using the Vitek 2 system.

Origin of Samples	Vitek Card Used			
	AST-GN27	AST-N093		
	*E. coli*	*E. coli*	*P. aeruginosa*	*P. putida*
Beef (*n* = 10)	6	-	2	2
Pork (*n* = 32)	26	2	2	2
Poultry (*n* = 10)	10	-	-	-
TOTAL (*n* = 52)	42	2	4	4

Legend: *n*—number of samples according to the origin of the carcass.

**Table 4 antibiotics-12-01408-t004:** Antimicrobial susceptibility profile of strains tested with the AST- GN27 card.

Antimicrobial			Susceptibility Test Result of 42 *E. coli* Strains (*n*/%)	
Class	Agent	MIC Range µL/mL	R	S
β-lactams	AMP	≥32	32 (76.1)	10 (23.8)
	AMC	8	26 (61.9)	16 (38.1)
	PIP	≥128	8 (19)	34 (80.9)
	TZP	≤4	-	42 (100)
cephalosporins	CFZ	≤4	30 (71.4)	12 (28.5)
	CXT	≤4	20 (47.6)	22 (52.3)
	CTX	≤1	4 (9.5)	38 (90.4)
	CAZ	≤1	2 (4.7)	40 (95.2)
	CPM	≤1	2 (4.7)	40 (95.2)
carbapenems	IMP	≤1	-	42 (100)
aminoglycosides	AMK	≤2	2 (4.7)	40 (95.2)
	GEN	≤1	2 (4.7)	40 (95.2)
fluoroquinolones	CIP	≤0.25	2 (4.7)	40 (95.2)
	NOR	2	8 (19)	34 (80.9)
tetracyclines	TET	≥16	16 (38.1)	26 (61.9)
nitrofuran derivative	NIT	≥16	22 (52.3)	20 (47.6)
sulfonamides	SXT	≤20	5 (11.9)	37 (88.1)

Legend: MIC—Minimum inhibitory concentrations, AMP—ampicillin, AMC—amoxicillin/clavulanic acid, PIP—piperacillin, TZP—piperacillin-tazobactam, CFZ—cefazolin, CXT—cefoxitin, CTX—cefotaxime, CAZ—ceftazidime, CPM—cefepime, IMP—imipenem, AMK—amikacin, GEN—gentamicin, CIP—ciprofloxacin, NOR—norfloxacin, TET—tetracycline, NIT—nitrofurantoin, SXT—trimethoprim-sulfamethoxazole, *n* = number of strains that expressed susceptibility (S) or resistance (R).

**Table 5 antibiotics-12-01408-t005:** Antimicrobial susceptibility profile of strains tested with the AST- N093 card.

Antimicrobial			Susceptibility Test Result of 10 Strains (*n*/%)					
Class	Agent	MIC Range µL/mL	*P. aeruginosa*		*P. putida*		*E. coli*	
			R	S	R	S	R	S
β-lactams	PIP	32	-	4		4	2	-
	TZP	32		4	2	2	-	2
	TIC	16	4	-	4	-	2	-
	TIM	16	4	-	4	-	2	-
cephalosporins	CAZ	4	-	4		4	2	-
	CPM	2	-	4	-	4	2	-
carbapenems	IMP	4	-	4	-	4	-	2
	MEM	≤0.25	-	4	-	4	-	2
aminoglycosides	AMK	8	-	4	-	4	-	2
	GEN	2	-	4	-	4	-	2
	ISP	8	-	4	-	4	-	2
	TOB	≤1	-	4	-	4	-	2
fluoroquinolones	CIP	≤0.25	-	4	-	4	-	2
	PEF	1	N.A.		N.A.		2	-
tetracyclines	MIN	4	N.A.		N.A.		-	2
sulfonamides	SXT	≥320	N.A.		N.A.		2	-
monobactams	AZM	16	4	-	4	-	2	-
polymyxins	COL	≤0.5	-	4	-	4	-	2

Legend: MIC—Minimum inhibitory concentrations, PIP—piperacillin, TZP—piperacillin-tazobactam, TIC—ticarcillin, TIM—ticarcillin/clavulanic acid, CAZ—ceftazidime, CPM—cefepime, IMP—imipenem, MEM—meropenem, AMK—amikacin, GEN—gentamicin, ISP—isepamicin, TOB—tobramycin, CIP—ciprofloxacin, PEF—pefloxacin, MIN—minocycline, SXT—trimethoprim-sulfamethoxazole, AZM—aztreonam, COL—colistin, n = number of strains that expressed susceptibility (S) or resistant (R), N.A.—not applicable.

**Table 6 antibiotics-12-01408-t006:** Antimicrobial resistance profile of the isolated *E. coli*.

Crt. No.	Origin	No. of Isolates	Vitek Card Used	No. of Classes with Resistance	Resistance to Antimicrobial Profile	Classes with Resistance
1.	Pork	2	AST-GN27	4	AMP, AMC, PIP, CFZ, CXT, CTX, CAZ, CPM, NOR, NIT	β-lactams, cephalosporins, fluoroquinolones, nitrofuran derivative
2.	Pork	2	AST-N093	5	TIC, TIM, PIP, CAZ, CPM, PEF, SXT, AZM	β-lactams, cephalosporins, fluoroquinolones, sulfonamides, monobactams
3.	Pork	2	AST-GN27	4	AMP, AMC, CFZ, CXT, CTX, NIT, SXT	β-lactams, cephalosporins, nitrofuran derivative, sulfonamides
4.	Pork	2	AST-GN27	5	AMP, PIP, CFZ, NOR, TET, SXT	β-lactams, cephalosporins, nitrofuran derivatives, tetracyclines, sulfonamides
5.	Pork	2	AST-GN27	4	AMP, AMC, CFZ, CXT, TET, NIT	β-lactams, cephalosporins, nitrofuran derivative, tetracyclines
6.	Beef	2				
7.	Pork	2	AST-GN27	3	AMP, AMC, CFZ, CXT, AMK, GEN	β-lactam, cephalosporins, aminoglycosides
8.	Pork	4	AST-GN27	3	AMP, AMC, CFZ, CXT, NIT	β-lactams, cephalosporins, nitrofuran derivative
9.	Beef	2				
10.	Poultry	2				
11.	Poultry	2	AST-GN27	3	AMP, PIP, CIP, NOR, TET	β-lactams, fluoroquinolones, tetracyclines
12.	Poultry	4	AST-GN27	3	AMP, AMC, CFZ, NIT	β-lactams, cephalosporins, nitrofuran derivative
13.	Pork	2	AST-GN27	3	AMP, PIP, NOR, TET	β-lactams, nitrofuran derivatives, tetracyclines
14.	Poultry	2				

Legend: Crt—current, No—number of samples, AMP—ampicillin, AMC—amoxicillin/clavulanic acid, PIP—piperacillin, CFZ—cefazolin, CXT—cefoxitin, CTX—cefotaxime, CAZ—ceftazidime, CPM—cefepime, NOR—norfloxacin, NIT—nitrofurantoin, TIC—ticarcillin, TIM—ticarcillin/clavulanic acid, PEF- pefloxacin, SXT—trimethoprim-sulfamethoxazole, AZM—aztreonam, TET—tetracycline, AMK—amikacin, GEN—gentamicin, CIP—ciprofloxacin.

## Data Availability

Not applicable.

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
