# Peer review of "Evaluation of Antimicrobial Resistance Profiles of Bacteria Isolated from Biofilm in Meat Processing Units"

_antibiotics, 2023, doi:10.3390/antibiotics12091408_

Round 1
Reviewer 1 Report
The manuscript entitled 'Evaluation of Antimicrobial Resistance Profiles of Bacteria Isolated from Biofilm in Meat Processing Units' by Ban-Cucerzan et al. regards the assessment of antimicrobial profiles of bacterial strains isolated from carcasses of different species. The study focused primarily on isolates of E.coli which resulted to be resistant to various antibiotics.
The manuscript is written in a chaotic way and it is really hard to follow. The Introduction section provides information without meaning and it does not guide the reader to the main point of it, which should be the problem that need to be addressed and the reason why the study has been performed in the first place. The M&M section is also poorly described. For instance, it is not clear the reason why the authors have chosen carcasses of different species and how they performed the sampling. The Results section is fairly well-written, but in the Discussion section, the authors have failed to provide any insights about their findings.
I believe this manuscript should be rejected in its present form.
Regards.
Reviewer 2 Report
The manuscript entitled "Evaluation of Antimicrobial Resistance Profiles of Bacteria Isolated From Biofilm in Meat Processing Units" aimed to analyze the biofilm formation on carcasses kept in cooling chambers for at least three days as well as to identify the bacteria developing in the biofilm and evaluate their phenotypic antibiotic susceptibility. The study is very important from a public health viewpoint. However, some issues needed to be clear, particularly, the methodology and the result sections.
Table 3- 42 isolates were tested for antibiogram. What is the source of these 42 isolates? And which bacteria they were? Please clear the issue. This is not mentioned in methodology too.
Table 4- Similarly in Table 4, What is the source of these 10 isolates? Pseudomonas spp. was tested. However, they isolated P. aeruginosa and P. putida. Which one was used for the antibiogram study? In addition, total resistance strain is not acceptable as they are different bacteria and please change the statements based on this result throughout the manuscript.
Please make all the tables and figures self-explanatory. Add notes for the short form used.
Please add a conclusion sentence in the abstract.
The justification of the study should be stronger. Why the study is different or what is new in this study should be mentioned. The discussion section is short. The Authors need to discuss the contamination and meat quality based on the current findings. Moreover, in most cases, they mentioned their findings and compared them with other studies only. They should add the cause of the findings in the discussion.
Line no. 165-167: The Authors mentioned that “In the present study the isolates expressed the highest resistance to ampicillin (AMP) and amoxicillin/clavulanic acid (AMC), these antimicrobials are commonly used in our country”. Please add the tested bacteria here and check the whole manuscript for similar corrections.
Line no. 207: What type of polythene bag was used? Any brand name? How did the Authors confirm the sterility of the bag?
Line no. 222-225: The Authors visualized biofilm under the microscope. How many samples were checked? Nothing was mentioned in the results section. Please add the microscopic view of the biofilm.
Line no. 303: Please add the full meaning of AMR.
Minor English corrections are required throughout the manuscript.
Reviewer 3 Report
This work analyzes the formation of biofilms on carcasses kept in cooling chambers, identifies the bacteria present in the biofilms, and evaluates their susceptibility to antibiotics. Overall, the manuscript provides valuable information.
However, some key results are missing, particularly related to the identification of biofilms. This is crucial for supporting the findings of this study, and the authors should supplement these results to strengthen their research.
Some major points have been given below:
1. The method section describes the identification of biofilms, but the corresponding results are not presented. The authors should provide this data to demonstrate the presence of bacteria in a biofilm state.
2. How many strains were identified in this study? Were there 52 isolates? Please provide a clear explanation of the strains included.
3. The authors used two types of antibiotic plates. Please explain the reason for this approach.
4. There are multiple spelling errors in the text and tables. Please check and revise them.
5. Lines 14-15: Please replace “an” with “and”. Also, when first mentioning E. coli and P. aeruginosa, please write out their full names.
6. Lines 15-16: Please rephrase this sentence for better clarity.
7. Line 17: The number of analyzed strains described in the abstract and results is inconsistent. Please ensure consistency and accuracy here.
8. The important conclusions and the research significance or value of this study are missing in the abstract. Please add these aspects to the abstract.
9. Please check and ensure consistency in units of temperature and time throughout the manuscript.
10. The discussion section should focus more on interpreting the results rather than simply comparing them with previous references. Please provide deeper analysis and insights into the findings.
11. The reference numbers in the text should be cited in the order of appearance.
Moderate editing of English language required
Round 2
Reviewer 1 Report
Accept in its present form.
Author Response
Dear reviewer, thank you very much for your overall positive feedback about the quality of our submission, and your valuable comments helping us to improve the quality of the manuscript.
Reviewer 2 Report
I would like to thank the Authors for their efforts. However, still few corrections required.
Table 2 is very confusing. It will create confusion for the readers. The Authors have mentioned that they have isolated 44 E. coli but in table 2, it is 40, same as for other bacteria. I found no use in Table 2. Please construct a new table showing how 52 isolates of different bacteria were obtained from the samples. It will be like Table 2 and it will show the final 52 isolates that were used for sensitivity test.
Line no. 186-187: Please check the sentence.
Moreover, please check the whole manuscript for similar errors.
Reference 45: Please check it. To me, it is not the appropriate citation. Cite the original reference work.
Minor English corrections required.
Reviewer 3 Report
I have no more questions.
Author Response

(The authors gave the same response as above.)
